



# Proglacial icings as records of winter hydrological processes

Anna Chesnokova[1], Michel Baraër[1], Émilie Bouchard[1]

[1]Construction engineering department, École de technologie superieure, Montréal, H3C 1K3, Canada

*Correspondence to*: Anna Chesnokova (chesnokovaanna@gmail.com)

**Abstract.** The ongoing warming of cold regions is affecting hydrological processes, causing deep changes such as a ubiquitous increase in river winter discharges. The drivers of this increase are not yet fully identified, mainly due to the lack of observations and field measurements in cold and remote environments. In order to provide new insights into the sources generating winter runoff, the present study explores the possibility to extract information from icings that form over the winter and are often still present early in the summer. Primary sources detection is performed using time lapse camera (TLC) images

of icings found in both proglacial fields and upper alpine meadows in June 2016 in two subarctic glacierized catchments in the upper part of the Duke watershed, St. Elias Mountains, Yukon. As TLC alone are not sufficient to entirely cover a large and hydrologically complex area, we explore the possibility to compensate that limit by four supplementary methods based on natural tracers: (a) stable water isotopes, (b) water ionic content, (c) dissolved organic carbon (DOC), and (d) cryogenic precipitates. Interpretation of the combined results shows a complex hydrological system where multiple sources contribute to

icings growth over the studied winter. Glaciers of all sizes, directly or through the aquifer, represent the major parent water source for icings formation in the studied proglacial areas. Groundwater-fed hillslope tributaries, possibly connected to suprapermafrost layers, make up the other detectable sources in icing remnants. If confirmed in other cold regions, those results will suggest orienting winter flow trend studies toward a multi-causal hypothesis in glacierized catchments. More generally, this study shows the potential of using icing formations as a new, barely explored source of information on cold regions' winter

hydrological processes that can contribute to overcoming the paucity of observations in these regions.

## 1 Introduction

Winter baseflow increases in response to climatic changes have been observed in a multitude of arctic and subarctic rivers (Brabets & Walvoord, 2009; Chesnokova et al., 2020; Danilovich et al., 2019; Jacques et al., 2009; Lammers et al., 2001; Rennermalm et al., 2010; Smith et al., 2007; Qin et al., 2020; Walvoord & Striegl, 2007; Wang, 2019; Woo & Thorne, 2014;

Yang et al., 2002). The hypotheses proposed to explain this positive trend mainly fall into two categories: those based on a change in water storage capacity of aquifers and those based on a change in water inputs to aquifers from different hydrological sources (Liljedahl, Gaedeke, O'Neel, Gatesman., & Douglas, 2016). Increases in aquifer storage capacity result from delays in soil freeze-up (e.g., Yang et al., 2002) and/or from increases in suprapermafrost layer thickness (e.g., Ge et al., 2011; Toohey et al., 2016). Those phenomena may lead to an increased amount of water accumulated in the aquifer during summer that can



be released during the freezing season. The term "suprapermafrost layer," as suggested by Connon et al. (2018), refers to the layer on top of permafrost and includes both the active layer and suprapermafrost taliks (i.e., perennially unfrozen zones). Hydrological sources that might increase their contribution to winter discharge include precipitation (e.g., Neal et al., 2002), permafrost thaw (e.g., St. Jacques et al., 2009), and glacier melt (Liljedahl et al., 2016). By increasing the aquifer recharge, those sources would indirectly promote an increase in winter discharge.

Although the aforementioned hypotheses are widely accepted, the exact sources responsible for the observed increase in winter discharge remain difficult to identify. This is especially true in proglacial settings where numerous hydrological components exist. Several studies have shown that the extra water volumes originating from the permafrost thaw alone cannot explain the observed trend in winter baseflow (e.g., McClelland et al., 2004). In addition, positive trends in winter discharge are observed in both permafrost- and non–permafrost-underlined areas (McClelland et al., 2004; Smith et al., 2007), and some

watersheds experienced a decrease in winter flow even though permafrost thaw had been observed (Lyon et al., 2009). Similarly, the hypothesis that glacier downwasting alone would be responsible for the changes in baseflow is confronted by the observed baseflow increase in both glacierized and non-glacierized watersheds (Chesnokova et al., 2020). Finally, due to data scarcity in cold regions, verifying the hypothesis about changes in precipitation input to aquifer recharge is problematic (Smith et al., 2007; Woo & Thorne, 2014). More generally, due to an overall lack of field observations over the winter, direct

evaluation of winter discharge composition in proglacial areas represents a major challenge.

        In the present study, we propose to compensate for the scarcity of direct observations of winter hydrological processes in glacierized catchments in cold regions by extracting information recorded in proglacial icings. An icing ("aufeis" in German, or "naled" in Russian) is a mass of ice formed during the winter when surface water or groundwater floods onto existing surfaces (ground, ice, snow) and freezes to form an additional ice layer (Carey, 1973; Kane, 1981). This flooding is intermittent

and may incorporate snowfalls (Hodgkins et al., 2004; Wadham et al., 2000). Icings redistribute runoff in the course of a year by releasing, during the ablation season, the water stored during the winter (Pavelsky & Zarnetske, 2017). Their contribution to summer runoff can be important for unglaciated watersheds in northern regions, accounting for up to 30% of the annual baseflow (Kane & Slaughter, 1973; Reedyk et al., 1995; Yoshikawa et al., 2007). In glacierized watersheds, icings can form from the different water sources that are active during the winter. Both polythermal (Bukowska-Jania & Szafraniec, 2005;

Hambrey, 1984; Sobota, 2016; Stachnik et al., 2016; Wadham et al., 2000; Yde et al., 2012) and cold-based glacier meltwater (Bælum & Benn, 2011; Hodgkins et al., 2004; Hodgkins et al., 1998; Naegeli et al., 2014) can be responsible for icing formation. At some locations, such features as lakes (Moorman, 2003; Veillette & Thomas, 1979; Wainstein et al., 2014; Yde & Knudsen, 2005) and buried ice formations within proglacial fields (Gokhman, 1987) can contribute to icing growth. In regions underlain by permafrost, water that flows to the surface through taliks over the winter has been observed being involved

in icing formation. Where this happens, both supra- (Ensom et al., 2020; French & Heginbottom, 1983; Pollard, 2005) and subpermafrost water (Ensom et al., 2020; Hu & Pollard, 1997; Kane & Slaughter, 1973; Kane, 1981; Pollard, 2005; Yoshikawa et al., 2007) can contribute.



Because types of water sources that produce proglacial icings are likely to contribute to winter runoff, learning from the hydrological records preserved in icings can provide new insights into winter hydrological processes and active sources in

many cold regions (Crites, Kokelj, & Lacelle, 2020). In light of the above, the main objective of this study is to explore the possibility to identify water sources that remain active during the winter in the headwaters of subarctic glacierized watersheds by tracing back those that contribute to the formation of icings. This is achieved by addressing three research questions: (1) What are the main parent sources responsible for icing formations in the studied watershed? (2) What is the most promising set of methods for extracting information from icings about winter hydrological process? And (3) what are the learnings on

winter hydrological processes and baseflow trends that can be derived from studying icing formations in the study area?

The Duke River valley, Yukon, Canada, is used as the study site. Numerous icings are observed almost every year in the upper part of the Duke watershed. In addition, this subarctic glacierized watershed is a complex hydrological system in which numerous hydrological components, such as glaciers, buried ice formations, talus slope and hillslope tributaries, snow cover, and permafrost, act as potential water sources. The study targeted six different icings found in two glacierized sub-

catchments in the upper part of the Duke watershed in June 2016 within both proglacial fields and upper alpine meadows.

Being the only direct method available for the study, the use of TLC images is considered as the primary detection method for parent hydrological sources for icing formation. Because using TLC at the scale of an entire sub-catchment is difficult and because TLC images may not detect some of the contributing sources, we tested four secondary methods in parallel: analyses of (a) stable water isotopes, (b) water ionic content (c), dissolved organic carbon (DOC) and (d) cryogenic

precipitates. As none of those four secondary techniques can be considered conclusive by themselves, the results they generate are ultimately expressed as "supporting" or "not supporting" a source as potential parent water for icing formation.

## 2 Study site

The two studied sub-catchments are located in the headwaters of the Duke River (Shar Ndu Chu) in the St. Elias Mountains, Yukon. The climate in this region is continental, with a mean annual air temperature varying between −2 and −6°C and a mean

annual total precipitation between 250 and 400 mm (Wahl, Fraser, Harvey, & Maxwell, 1987). The glaciers in the region are not in equilibrium with the current climate and experience long-term negative mass balances (Arendt et al., 2002; Barrand & Sharp, 2010). Large glaciers are polythermal (Flowers, Copland, & Schoof, 2014), whereas smaller glaciers are probably cold-based (Wilson, Flowers, & Mingo, 2013). Many glaciers in the area, including the Duke Glacier, are surging (Clarke et al., 1986). As a result of glacier retreat and surging events, their proglacial fields contain climate-sensitive hydrological features

such as buried ice and ice-cored moraines (Johnson, 1971; 1978; 1986; 1992).

The study region is located in a zone of discontinuous permafrost (Brown, Ferrians, Heginbottom, & Melnikov, 2002). However, local permafrost coverage can be continuous at elevations above 1600 m above sea level (a.s.l.) (Bonnaventure & Lewkowicz, 2008). Most of the permafrost monitoring sites in Northern America display a thawing trend driven by air



temperature increase. Indeed, permafrost warming and active layer deepening are reported in the arctic (Hinzman et al., 2005)
and the subarctic (Smith et al., 2010; Smith et al., 2005; Tarnocai et al., 2004).

The two study catchments are contiguous. As we were unable to learn the name of that particular valley, we refer to the first study catchment as Watershed B (Figure 1a). Its drainage area of 8.7 km$^2$ is 36.6% glacierized, and its elevation ranges from 1674 to 2906 m a.s.l. The Watershed B surface is, to a large extent, composed of bare rocks and debris. The main stream originates from a glacier, which we named B. The other two glaciers in the same Watershed B form a tributary that reach the
main stream in its upper section.

The second study catchment is named Upper Duke watershed (Figure 1a). Its drainage area of 69.1 km$^2$ is 39.7% glacierized, and its elevation ranges from 1598 to 3042 m a.s.l. The main glacier of the catchment is named Duke Glacier. It is categorized as a surging glacier (Clarke et al., 1986). The Duke Glacier is considered the Duke River headwater. The Duke River is fed by numerous tributaries of glacier and non-glacier origins as it flows downstream. This part of the watershed can
be divided into two areas of different natures (Figure 1a): starting around the actual Duke Glacier tongue and extending several kilometers downstream is the proglacial field. It is a barely vegetated debris-covered area that has been shaped by previous glaciations and/or surging events. Further down, the watershed is characterized by alpine meadow with less pronounced vegetated slopes.

## 3 Methods

### 3.1 Methods overview

The analysis of TLC images, in conjunction with the winter air temperature records, is conducted for two winters: 2015-2016 and 2016-2017. This primary method permits the monitoring of icings formation as well as visual location of hydrological sources contributing to their growth throughout winter. The TLC images analysis presents an advantage over other methods since it is based on direct observations with limited possibilities of producing false-positive results (i.e., indicate sources which
are not active). TLC coverage, however, is limited in its range of coverage, vulnerable to certain weather/environmental conditions (e.g., snowfall, darkness) and is inconclusive for parent sources that are not observed on the images. We thus test four other methods which potentially are able to provide complementary information on parent sources responsible for icing formation. Those indirect methods are all based on the principle that waters originating from different sources present unique hydrochemical and isotopic signatures as a result of the specific hydrological, geological, and biological processes they have
been exposed to (Drever, 1997). We hypothesize that comparing the hydrochemical signatures of icings water and cryogenic precipitates (solid samples) to those of other hydrological sources can provide insights about the sources that contributed to icing formation.

Taken individually, each of the indirect methods presents a substantial level of uncertainty. They are all based on the hypothesis that water chemical and isotopic properties of parent sources do not change drastically over the year in glacierized
catchments, as was observed by Beylich and Laute (2012). This hypothesis can be considered robust in the case of DOC and



cryogenic precipitates. In pristine environments, DOC originates mainly from plant and microorganism decomposition (Hagedorn, Saurer, & Blaser, 2004). The presence of significant amounts of DOC in icings can therefore allow rejection of the contribution from DOC-free parent sources. Similar reasoning is used for cryogenic precipitates. The formation of cryogenic precipitates requires high relative concentrations of the elements that precipitate in parent waters (Bukowska-Jania, 2007) that

further increase due to freezing-driven exclusion (Kuo, Moussa, & McNeill, 2011). The same phenomenon is susceptible to affecting the relation between the relative concentration of solutes in parent sources and in icing meltwater. In addition, because there is a link that exists between stable water isotopes and elevation (Baraer et al., 2015), isotopic signatures of meltwater can potentially vary throughout the year. Moreover, icings' remnants, which are observed in spring, might not be formed by a unique source/process, but rather may represent aggregated features formed as a result of multiple sources and processes.

Taking into account those limits, interpretation of the results for indirect methods for each potential parent source is simply expressed either as (+) indicating that the source is supported as a contributor to the icing formation, or as (-) indicating that the source is suggested as not being a contributor. Resulting outcomes are compiled into a decision matrix. Unlike the indirect methods, sources identified as contributing to the growth of a given icing are given a (++) as TLC images interpretation is considered the primary method. Due to the limitations mentioned above, the analysis of stable water isotopes, DOC and

cryogenic precipitates were used only to rule out potential parent sources (i.e., support as non-contributor). Those are based on the principle that similitudes between icing and a given source for a unique tracer do not necessarily imply the source is a parent water, but dissimilitude suggests the source is not a parent water. The opposite principle applies where many tracers are used, as is the case for the ionic signature-based PCA.

The methodological approach is first applied to Watershed B, which is smaller and of lower hydrological complexity than

the Upper Duke. The Watershed B serves as a proof of concept and we use it to fix or validate thresholds for secondary methods such as the analysis of stable water isotopes. Once tested on Watershed B the methods are then applied to the hydrologically complex Upper Duke watershed.

### 3.2 Analysis of time lapse images

TLCs were oriented to target icings formation with the objective of capturing flooding events and possible contributing sources.

We installed three TLCs during summer 2015 (Figure 1a). TLC1 was installed above Glacier B's tongue and pointed toward the Glacier B proglacial field, proglacial icing (Icing PF), and the right-side moraines. TLC2 was installed in the Duke watershed, just above the Duke Glacier margin, and pointed toward the Duke Glacier's proglacial field and the Duke River, where Icing 4 formed. TLC3 was placed just above the proglacial field–alpine meadow border and pointed toward the Duke Glacier's tongue and the upper part of its proglacial field, thus featuring Icings 3 and 4. All cameras took four visible color

images per day at 8h, 11h, 13h, and 16h at 72 dpi.

Visual analysis of TLC images consisted of identifying the timing and the extent of flooding events and snowfalls. On TLC images, flooding events are associated with a darkening/bluing of the white ice/snow pixels due to increased water content (e.g., Figure 2b–d) or with a sudden appearance of bright pixels due to the formation of a reflecting layer on top of a previously



low-reflecting one (e.g., Figure 3b and f). Major snowfall events erase traces of flooding on images by whitening the scene,
which facilitates detection of subsequent floods. TLC time series are occasionally interrupted when snowfalls cover the lenses
of the cameras. For each detected flooding event, we listed possible water sources based on the flooding location. The timeline
of the detected flooding events was then superimposed on the air temperature time series of an automatic weather station
installed on the right shore of the Duke River upstream from the confluence with Stream B (Figure 1a). This station measured
mean hourly air temperatures from 11 July 2015 to 24 June 2017 at a height of 1.5 m above the ground.

### 3.3 Sample collection

During the field campaign of June 2016, we observed and sampled four icing remnants within the Duke riverbed (Figure 1a
and c) and two in Watershed B (Figure 1a and b). Two of the icings in the Upper Duke watershed (Icings 3 and 4) were situated
within the proglacial field, and the two others (Icings 1 and 2) within the alpine meadow (Figure 1a and c). Icing varied from
70 to 1980 m in length and from several meters to several hundreds of meters in width. Ice thickness was estimated to range
from 0.5 m to 1.5 m. In Watershed B, Icing B stretched along the river and was 60 m long and 10 m wide. The second icing
(Icing PF) was situated within the proglacial field next to Glacier B's tongue and had a circular shape, ca 40 m in diameter
(Figure 1a and b). All icings were characterized by an alternating sequence of layers of blue bubble-free and white bubble-
reach ice, which result from slow and quick freezing of water, respectively (Moorman & Michel, 2000). Icings 1 and 2 had
layers of candle ice closer to their tops, suggesting that melt events had already altered the ice (Stachnik et al., 2016). An ice
mound, a feature that forms when the top surface layer bulges due to the freezing of water lenses trapped underneath (Carey,
1973), was still intact on Icing 4. Overall, 13 icing samples were taken in June 2016 (Figure 1b, c, Table 1), from supra-icing
ponds/channels (samples 1.3, 2, 3, 4.2, 4.4, Icing B, and Icing PF), from water dripping from the icing surface (samples 1.2,
4.1, 4.3), or in the form of ice crystals from one of the icing layers (sample 1.1).

Over the same period, we sampled all potential contributors to icing formation (Table 1). To make the distinction
between the roles of proglacial fields and of alpine meadows in icing formation, samples of potential contributors were
categorized as PF or AM respectively.

In Watershed B, the main glacier was sampled at the edge of its immediate proglacial field. Both left and right lateral
ice-cored moraines were sampled from purged boreholes made in the buried ice with a steam drill. Within the left moraine
complex, we also sampled a moraine lake situated above the main stream but not visibly connected to it. In the downstream
part of the watershed, hillslope tributaries originating from taluses were sampled close to their confluence with the main stream.
Finally, we sampled a large tributary fed by the two other glaciers of the same watershed (Figure 1a).

In the Duke watershed proglacial field, Duke Glacier was sampled from water discharging at the glacier snout. Samples
from buried ice formations and kettle lakes were found in the only proglacial field. Another proglacial field-specific group of
sources includes small tributaries with headwaters within the borders of proglacial field, called PFL tributaries.
Alpine meadow-specific sources consisted of groundwater and suprapermafrost layer water. The right side of the alpine
meadow exhibited permafrost-specific features such as thaw scars. Several small disconnected ponds/springs were sampled as





suprapermafrost layer water, even though they may include a limited amount of direct precipitation. In this area, digging shallow wells has been impossible due to the presence of ice in the ground. No visible signs of permafrost were observed at the left side of the alpine meadow. Samples taken from the wells on that side, once purged, were therefore identified as

groundwater. We acknowledge that these samples may also represent water from the suprapermafrost layer, but we decided make a distinction between them due to their different sampling locations.

Hillslope tributaries from glacier-free areas (with the headwaters on the valley hillslopes) and streams fed by glaciers were found within both the proglacial field and the alpine meadow (Figure 1a). The latter were sampled twice, upstream close to the glacier snout and downstream close to the confluence with the Duke River. Finally, we also collected rainwater samples

during the field campaign.

All the water samples were taken in a very short period of time following a synoptic sampling strategy (Mark & Seltzer, 2003). Samples were collected into high-density *polyethylene* plastic bottles. We collected 30 mL for stable water isotope analysis and 50 mL for solutes and DOC analyses. Samples for solutes and DOC were filtered using 0.45-micron filters and acidified with three drops of nitric acid 0.5M in the field. All bottles were filled to the brim, sealed, and kept in the dark at 4°C

prior to analysis.

During the 2016 field campaign, solid state samples were collected on Icings 1 and 3 in the Duke watershed, and on Icing B in Watershed B (Figure 1b and c; Table 1). These samples were assumed to be cryogenic precipitates. Cryogenic precipitate is commonly encountered on the icing surface (Lacelle, Lauriol, & Clark, 2009). It is formed when sufficient concentrations of precipitating elements exist in the freezing water (Žák, Onac, & Perşoiu, 2008). The production of mineral precipitates in

such conditions occurs by solute expulsion during freezing (Bukowska-Jania, 2007; Lacelle et al., 2009; Lauriol et al., 1991; Vogt, 1991). The most common minerals found at the surface of the icings in the melt season are carbonates ($CaCO_3$) and gypsum ($CaSO_4$) (Lacelle et al., 2009). In glacierized catchments, those cryogenic minerals represent a unique archive of the hydrologic conditions that prevailed at their formation (Thomazo et al., 2017). We collected solid state samples where clear association between the precipitate and the icing was possible. Around 5 g of samples were taken and stored into sealed 50

mL plastic centrifuge tubes.

**3.4 Analysis of stable water isotopes**

We analyzed water samples for relative concentrations of stable isotopes of oxygen, $\delta^{18}O$, and hydrogen, $\delta D$, using cavity ring-down spectrometry (Picarro Analyzer L2130-I; guaranteed instrumental precision is 0.03‰ for $\delta^{18}O$ and 0.2 ‰ for $\delta D$). Both $\delta^{18}O$ and $\delta D$ were expressed in per mil relative to Vienna Standard Mean Ocean Water standard (‰) (Coplen, 1996). The

spectrometer was calibrated every 100 samples using six laboratory standards. In addition, we analyzed one standard after every third sample to verify the stability of measurements and eventually perform corrections. Results are presented in the form of $\delta^{18}O$-$\delta D$ plots, which include Local Meteoric Water Line (LMWL) built based on rain samples. Potential sources which plot away from the icing samples are then considered as non-contributing sources. The thresholds applied for this decision are fixed using Watershed B results as a reference.





### 3.5 Analysis of selected ions

We measured the anions concentration in the samples ($F^-$, $Cl^-$ and $SO_4^{2-}$) using an ionic chromatographer (Dionex ED50, Thermo Fisher Scientific) that was calibrated every 30 samples using seven standards ranging from 0.005 to 3.2 ppm for $F^-$, from 0.1 to 50 for $Cl^-$, and from 0.2 to 75 for $SO_4^{2-}$. We inserted a standard (0.2 ppm for $F^-$, 0.9 for $Cl^-$ and 1.8 for $SO_4^{2-}$) followed by a blank every three samples to guarantee analytical stability and to correct results from an eventual drift in measurements.

Cationic concentrations ($Li^+$, $Na^+$, $K^+$, $Ca^{2+}$, $Mg^{2+}$, $Al^{3+}$, $Ag^+$, $Ba^{2+}$, $Cr^{3+}$, $Cu^{2+}$, $Fe^{3+}$, $Mn^{3+}$, $Si^{4+}$, $Sr^{2+}$, $Ti^{3+}$, $Zn^{2+}$) were measured using an inductively coupled plasma optical emission spectrometer (ICP-OES, 5110 Agilent). Calibration was realized every 23 samples using nine standards ranging from 1 to 40 ppm for $Ca^{2+}$, $Si^{4+}$, and $Mg^{2+}$ for some samples (those with high concentrations), and from 0 to 10 ppm for the other ions. We inserted two standards (10 ppm for $Ca^{2+}$ and 5 ppm for all other ions) followed by a blank sample every three samples to guarantee analytical stability and to correct results resulting from a drift in measurements.

We investigated connections between samples by conducting a principal component analysis (PCA) on selected tracers. Tracers were selected based on their ability to segregate different sample origins/types in bivariate plots (Baraer et al., 2015). As the hydrochemical signature of the icing samples depends not only on the hydrochemical composition of its parent sources, but also on the cryochemical fractionation, we used the relative concentrations of solutes (ion concentrations divided by the sum of anions/cations) instead of absolute concentrations in the PCA (Baraer et al., 2015). For each group of hydrological sources, we added error ellipses that illustrate a 95% confidence interval assuming Gaussian distribution. After being tested on the Watershed B dataset, PCA diagrams were used in a way that contribution of a given group of hydrological sources to icing formation is supported if icing sample plots are within the error ellipse of this group.

Alongside PCA, we created conceptual maps of dominant ions' relative concentrations to identify areas with particularly high concentrations of a given solute. The upper quartile calculated for each selected ion based on its relative concentrations was chosen as a threshold value.

### 3.6 Analysis of dissolved organic carbon

Water sources originating from permafrost-underlined vegetated terrains are usually characterized by higher concentrations of DOC than the nearby permafrost-free areas due to extended exposure to the organic-reach suprapermafrost layer (Carey, 2003; Ma et al., 2019; MacLean et al., 1999). DOC can therefore be seen as a tracer for permafrost-related water sources (e.g., Petrone et al., 2006; Toohey et al., 2016; Yoshikawa et al., 2007). We measured DOC concentrations using an Apollo 9000 Combustion Analyzer. Three injections were made for each sample, and calibration was done every 20 samples using four standards ranging from 0 to 10 ppm. We analyzed a standard every five samples to assess the stability of the measurements. The detection limit for the analysis was 1 ppm. To avoid false-positive results, the threshold for considering a DOC presence




in a sample was fixed at twice the limit of detection. Hydrological sources with less than 2 ppm of DOC are therefore considered as non-contributing to those icings that exhibit DOC concentrations in their samples over this limit.

### 3.7 Analysis of solid samples

We applied two methods to determine the chemical composition of solid samples: X-ray fluorescence (XRF) (Thermo Scientific, Niton XL3t GOLDD) and acid digestion followed by anion and cation analysis. All samples were first dried at 110°C for 24 hours. For XRF, a smooth and plain pellet was produced out of 2 g of sample using a pellet press. For acid digestion, 1 g of dried sample was mixed with 4 mL of nitric acid (50%) and 10 mL of hydrochloric acid (20%) and heated for 2 hours at 90°C. The resulting solutions were then analyzed by ICP-OES (see above for the characteristics) for cationic concentrations. We employed both methods to determine the relative abundance of ions. Parent source identification was made 265 by comparing the chemical composition of the samples with the ionic concentrations of water samples from different sources by using the conceptual maps of selected ionic relative concentrations (Section 0). Hydrological sources that do not present precipitates constituents as a dominant element in their samples are considered as non-contributors to the icing from which cryogenic precipitates originate.

## 4 Results for Watershed B

### 4.1 Time lapse image analysis

In Watershed B, we detected episodes of hydrological activity leading to the formation of icing PF throughout both winter seasons. Icing B was not visible on TLC images. The images showed that the remnants of Icing PF persisted late into the melting season for both years (end of September 2016 and July 2017).

The first winter of the study (2015–2016) was characterized by consistent negative air temperatures between late 275 September and late March, except during two warm events: 25–27 November and 30 December. The first identified flooding event occurred in mid-October (Figure 4, top panel), and, by the beginning of November, we observed icing formation on the left side of the proglacial field. In general, flooding events over the icing surface occurred mainly in two locations: on the left side until January (e.g., Figure 2b), where the main glacier outlet stream is observed in the summer; and from January onwards most often on the right side of the proglacial field (e.g., Figure 2c). On two occasions during that season, 16 February and 6 280 March, flooding occurred next to the left-side lateral moraine, where buried ice is exposed in the summer. On 16 February, we observed a stream forming on the slope of this ice-cored moraine (e.g., Figure 2d and e). As a small moraine lake is situated at the top of this moraine, this stream could originate either from the lake or from buried ice.

During winter 2016–2017, the air temperature stayed below zero between mid-October and mid-April, apart from warm episodes on 30 December and 2 January (Figure 4). The icing's extent increased in late fall, when the air temperature dropped 285 below zero. By the end of October 2016, an icing had formed on the right side of the proglacial field (Figure 4, bottom panel). Unlike the 2015–2016 season, flooding events mostly happened on the right side of the proglacial field before January, and on





its left side afterwards. During the 2016–2017 season, we observed signs of contribution of either the left ice-cored moraine or the moraine lake to icing formation on 26 November and 28 March.

For both winters, the sources identified as potential contributors to icing growth based on TLC images are the main

glacier and either the left ice-cored moraine or the moraine lake, or both.

### 4.2 Analysis of stable water isotopes

On the TLC images, we observed that Glacier B contributes to the Icing PF formation (Figure 4). Based on this finding, we use the distance between samples from Icing PF and Glacier B on the $\delta^{18}O$– $\delta D$ diagram to form a range from other icing samples in both directions along the LMWL for decision making on isotopic compatibility between icings and potential parent

sources (red arrows on Figure 5a), where samples outside of the range are ruled out as potential parent sources. As a result, the sample from the moraine lake cannot be ruled out as a potential parent source for Icing PF since it plots within the range. The sample from the left ice-cored moraine plots further away from the Icing PF sample, and thus its contribution is not supported. Concerning Icing B, we observe that its sample plots together with those of the hillslope tributaries and away from the other tested sources.

### 4.3 Analysis of ionic signatures


The results of ionic concentration analyses are presented in Figure 5b, which shows the projection of water sample data on the reduced space determined by PCA using $Ca^{2+}/SC^+$, $Mg^{2+}/SC^+$, $Li^++Na^++K^+/SC^+$, $SO_4^{2-}/SA^-$, and $Sr^{2+}/SC^+$. We chose these tracers based on their ability to cluster icings and different water sources in bivariate plots (Figures S1 and S2, Supporting Information). For Watershed B the first two principal components account for 0.86 of the total variance. Due to the small

number of samples it was not possible to include error ellipses for any group of hydrological sources apart from hillslope tributaries. In the PCA diagram (Figure 5b), none of the sources plot close to the Icing PF. The sample from Icing B plots within the error ellipse of hillslope tributaries and thus analysis of ionic signatures supports the contribution from hillslope tributaries to Icing B formation. The use of the 95% confidence intervals thus was validated based on the fact that for Icing B, the method provides the same indications as stable water isotopes analysis.

### 4.4 Dissolved organic carbon


None of the samples taken in Watershed B presented a DOC concentration above the equipment detection limit, making this method non-conclusive for this watershed.

### 4. 5 Analysis of solid samples

The solid sample taken on Icing B was brown-grey and, according to both methods, namely XRF and acid digestion, presented

high concentrations of $Al^{3+}$ and $Fe^{3+}$ (Figure 6). Those elements do not correspond to reported compositions of cryogenic





precipitates occurring naturally on the earth (Lacelle et al., 2009). Thus, sampled solids most probably are sediments carried out by parent waters. Unlike for cryogenic precipitates, the relation of those sediments to icing formation cannot be proven. On the other hand, high relative concentrations of both $Al^{3+}$ and $Fe^{3+}$ are observed in the water sample from the moraine lake, and high relative concentrations of $Fe^{3+}$ are observed in the hillslope tributary sample s2 (Figure 6), making those sources

possible contributors to the formation of Icing B.

## 5 Results for the Duke watershed

### 5.1 Time lapse images analysis

In the Duke watershed, we detected hydrological activity only for Icings 3 and 4. In comparison with Watershed B, we identified a larger number of water sources as potential contributors to those icings' formation. In general, flooding detected

by TLC occurred when the air temperature was below zero, and all detected sources were active during both seasons. On occasion, several sources showed activity at the same time (on 29 November or 10 December, Figure 7 top panel).

In the 2015–2016 season, Icings 3 and 4 had formed by the end of November and melted away by the end of July. Due to TLC failure, observations for this season are only available until February. The hydrological activities of Duke Glacier and small glacier Gl2 were detected throughout the observation period (Figure 7, top panel). The only flooding event associated

with the small glacier Gl2 was visible in November. In addition to glaciers, hillslope tributaries s3-s7 appear contributing to icings growth. Tributaries s5, s6, and s7 are represented by the same circle in Figure 7 since distinguishing between them based on TLC images in winter conditions was not possible.

The 2016–2017 icings formed at the beginning of November. Icing 3 had melted away by the end of July, and Icing 4 persisted until the end of August (Figure 7, bottom panel). In November 2016, Icings 3 and 4 appear to make one larger icing.

The Duke Glacier and small glaciers Gl1 and Gl2 contribute to icing formation throughout this winter season (Figure 7, bottom panel). As for hillslope tributaries, s3 and s4 were active before January, and s5, s6, and s7 showed activity from January onwards (Figure 7, bottom panel).

For both winters, the sources identified as potential contributors to Icing 4 growth include the Duke Glacier, streams fed by two small glaciers on the right side of the valley, and two hillslope tributaries on the right side. For Icing 3, potential

contributors include three hillslope tributaries on the left side of the valley.

### 5.2 Analysis of stable water isotopes

Figure 8a and 8c present the results from stable water isotopes for the Duke watershed, where Figure 8a shows results for icings and potential sources within the proglacial field, and Figure 8c shows results for the alpine meadow. Samples from the most upstream icing, Icing 4, plot in the lower part of the $\delta^{18}O–\delta D$ diagram (Figure 8a). When we use the distance obtained

for Watershed B as a reference, we see that such hydrological sources as the main glacier, buried ice formations, PFL sources, and both glacier-fed tributaries and hillslope tributaries plot within the range indicated by red arrows for Icing 4 (Figure 8a)





and thus are not ruled out as potential parent sources. For Icing 3 this range includes PFL sources, glacier-fed tributaries, and hillslope tributaries. For alpine meadow Icing 2, glacier-fed and hillslope tributaries plot close to the icing sample, and for Icing 1, in addition to these two groups of sources, samples from groundwater wells also plot within the range formed by the
arrows (Figure 8d).

### 5.3 Analysis of ionic signatures

Figure 8b and 8d present the results from ionic concentration analyses for the Duke watershed, where the first two PCs account for 0.85 of variance. For readability, we split the results presentation into two sub-figures: Figure 8b shows results for icings and potential sources within the proglacial field; Figure 8d shows results for icings and potential sources within the alpine
meadow. On the PCA diagram, samples from both proglacial field icings, Icing 4 and 3, plot within the error ellipses formed by samples from buried ice formations (Figure 8b), and thus ionic signature analysis supports these formations as parent water sources. For Icing 3 this method also supports moraine lakes as a parent source (Figure 8b). Alpine meadow Icing 2 is potentially fed by hillslope tributaries and Icing 1 samples do not appear to be related to any water sources (Figure 8d). Note that the ionic signature of sample 1.1 was not analyzed since it was taken in the form of ice crystals.

### 5.4 Dissolved organic carbon

Most of the samples taken from icings have DOC concentrations below the fixed threshold (2 ppm), with two exceptions: sample 1.2 (Icing 1) presents DOC concentrations at 5.9 ppm, and sample 2 (Icing 2) at 2.3 ppm. Among potential parent waters, only the samples from the alpine meadow ponds and those taken from groundwater wells have high DOC concentrations: P1 (8.6 ppm), P2 (5.4 ppm), P3 (5.4 ppm), Gw1 (4.0 ppm), and Gw2 (4.3 ppm). The hillslope tributary sample
S13, sampled just upstream of the suprapermafrost ponds, also exhibit higher than the 2 ppm threshold DOC (2.6 ppm). DOC concentrations below the threshold in all other samples suggest that those should be considered non-contributing sources to icing samples 1.2 and 2. As in permafrost environments, DOC has been associated with suprapermafrost layers and because all samples presenting DOC in high enough concentrations are located in an area where permafrost has been detected, those results support the hypothesis of a suprapermafrost water contribution to icing in this lower
section of the upper Duke valley.

### 5.5 Analysis of cryogenic precipitate

The solid sample S3 from Icing 3 was white and presented high amounts of Ca and S when analyzed with XRF, and of $Ca^{2+}$ and $SO_4^{2-}$ upon acid digestion (Figure 6). Such composition is characteristic of cryogenic gypsum, the second most common cryogenic mineral found on icings (Lacelle et al., 2009). Figure 6 shows high relative concentrations of $Ca^{2+}$ in buried ice
samples (BI 2 and 3) and in samples from the suprapermafrost ponds (P1–3) on the right side of the alpine meadow. High relative concentrations of $SO_4^{2-}$ are observed throughout the proglacial field, apart from samples of hydrological components that are related to the Duke Glacier (i.e., samples from buried ice formations, from Duke Glacier, and on the left side of the





alpine meadow). All the above-mentioned sources can potentially reach gypsum saturation during freezing and solute expulsion, except the ones that also show high concentrations of carbonates: buried ice formations and suprapermafrost ponds

(Figure 6). Because the solubility constant of calcite is more than 1000 times lower than that of gypsum (Zarga, Ben Boubaker, Ghaffour, & Elfil, 2013), cryogenic precipitates from those sources should be made mainly of calcium carbonate. The results, therefore, do not support sources which have relatively high concentrations of carbonates as the parent water for Icing 3, including the Duke Glacier, since buried ice within the proglacial field is likely made of its remains.

The solid sample collected on Icing 1 was brownish-grey and presented high concentrations of $Ca^{2+}$ and carbonates

(Figure 6). Figure 6 shows high relative concentrations of carbonates and calcium in the upper part of the proglacial field in water samples from buried ice formations (BI2-5) and on the right side of the alpine meadow in samples of suprapermafrost layer water (P1–3) and hillslope tributaries s13 and s11. The distance from buried ice and those hillslope tributaries makes their contribution to Icing 1 formation highly improbable.

## 6 Discussion

### 6.1 Icing formations parent sources identification

### 6.1.1 Identification of parent sources based on TLC images

For each studied icing, Table 2 provides a summary of the sources contributing to icing formation based on each method, where only sources that can physically contribute (i.e., geographically close, not downstream of the icing) to each icing are listed. The results from our primary method, the TLC images, showed contributing water sources for each of the icings that

was in the cameras' field of vision. We marked those sources as "++" in Table 2. The main glaciers in both studied watersheds feed their most proximate icings, Icings PF and 4, and glacier-fed tributaries as well as hillslope tributaries contribute to Icings 3 and 4 formation. For Icing PF, another contributing source was identified, but images did not allow differentiation between a buried ice formation and a moraine lake. Thus, we marked both sources with a single "+". Those primary results highlight that multiple sources are contributing to the icings' growth, even for icings forming close to the glacier's tongue. This first

finding contrasts with the traditional characterization of icing based on a single contributing source (e.g., Åkerman, 1982; Crites et al., 2020). Those results show how important tributaries are in hydrologically complex set-ups, such as the Upper Duke River watershed. There are two types of contributing tributaries: connected to glaciers and not. The contribution to icing formation of small glaciers, which are situated on the main valley slopes, can be hypothesized to be either in the form of surface discharge, or discharge through the ground. The latter would mean that those small glaciers are an important source of

aquifer recharge. The contribution from non-glacier-fed hillslope tributaries was observed over long periods of subzero air temperatures (Figure 4Figure 7), and thus those tributaries can be considered as groundwater-fed.





### 6.1.2 Identification of parent sources based on indirect methods for icings within TLC field of view

We tested four indirect methods to detect parent sources because TLC do not cover all the observed icings, and because hydrological sources may be missed by this method. Noticeably, these indirect methods did not contradict any of the findings
from the primary method: none of the indirect methods suggested that the main glacier is not a parent source of the Icing PF, and that glacier-fed tributaries and hillslope tributaries are not parent sources of Icings 3 and 4. Apart from confirming sources identified by TLC, indirect methods also help detect potential parent sources which were not visible on TLC images. In Watershed B, stable water isotopes support moraine lakes as contributors to Icing PF, but not buried ice formation. Thus, indirect methods help to verify sources observed on TLC images for this icing. In the Duke watershed, for Icing 4, the
contribution of buried ice formations is supported by ionic signature analysis and is not ruled out by analysis of isotopes (Table 2). Regarding Icing 3, not much is brought by the indirect methods. For this icing, the results of ionic signature analysis and isotopic analysis disagree for such sources as buried ice formations and moraine lakes. In particular, buried ice formations are rejected as potential contributors by both analysis of solid samples and isotopes analyses, and thus we can consider their contribution to Icing 3 as of low probability. None of indirect methods supported main glacier contribution to Icing 3 formation.
This is in agreement with TLC images and suggests that TLCs did not miss any flooding events from the Duke Glacier.

### 6.1.3 Identification of parent sources based on indirect methods for icings that are not visible on TLCs

In Watershed B we observe a 100% agreement between the three methods that were applied to Icing B (Table 2), where all methods support hillslope tributaries (groundwater-fed) as its unique parent source. These results, however, should be met with caution as such method as analysis of isotopes has been parametrized based on the $\delta^{18}$O–$\delta$D diagram obtained for
Watershed B. In the Duke watershed, contribution of groundwater-fed hillslope tributaries to the formation of Icing 2 is supported by the analysis of ionic signatures and is not rejected by any other method, and main stream as a source is not supported by any method (Table 2). When we look at the results for other potential parent sources for Icing 2 and for any source for Icing 1, we see that there is no full agreement between the methods.

### 6.2 Methodological limits

The use of TLC image analysis as the primary method allowed us to capture major flooding events and identify hydrological sources responsible for those events with a reasonable level of confidence. There are, however, three main drawbacks associated with the TLC images: 1) the difficulties we found in differentiating between sources situated close to each other, 2) flooding events missed because of snowfall events as well as shorter daylight hours during the winter, and 3) the contribution of some minor sources was not detectable. In addition, the cameras' limited field of view and the difficulties in finding shooting
locations that allow image acquisition throughout the whole year proved to be challenging in complex environments. Finally, harsh winter conditions can be responsible for equipment malfunctions. For instance, due to technical problems, one of the cameras in the Duke watershed stopped working at the end of the 2015–2016 season, thus eliminating the possibility of



comparing the number of hydrological events for each identified parent source between seasons (Figure 7, top panel). The use of indirect methods, which are based on natural tracers, to support TLC image interpretation was informative in the studied
watersheds as none of those methods contradicted TLC findings. When applied to icings which were not covered by a TLC, those indirect methods showed contrasting results. In the smaller Watershed B, they exhibited good agreement for Icing B, providing a good confidence level in the overall findings. The situation is less straightforward for Icings 1 and 2 in the Duke watershed. From a total of 11 potential sources evaluated for those two icings, the full agreement between applied indirect methods was reached only twice (Table 2). This lack of uniformity between the results of different methods partly arises from
the underlying assumptions and sampling procedure. All methods that use naturel tracers are based on assumptions about the tracer's content conservation across the seasons, which can rarely be verified. While this assumption is conceptually strong for cryogenic precipitates and DOC, it is less certain for stable water isotopes and ionic signatures. We can only use analysis of cryogenic precipitates and DOC where those elements were sampled/detected in icing samples. Thus, the results from these analyses have a lower weight in the decision matrix presented in Table 2 compared to results from the isotopes and the ionic
signature analyses. In addition, the four indirect methods used are highly dependent on the sampling strategy. This is illustrated by two icings that have been sampled at multiple points, such as Icings 4 and 1: both icings show a variability in isotopic and ionic signatures among their samples.

Finally, while the use of indirect methods helped to account for the spatial variability of hydrological components responsible for icings formation, these methods did not provide information about temporal variability of hydrological activity.
Here, despite being unable to give a general picture of sources that remain active all winter long, analysis of solids and DOC helped to provide indications about parent sources in specific cases such as the contribution of the suprapermafrost layer water.

**6.3 Learnings on winter hydrological processes and implications for baseflow trends in the Upper Duke River valley**

All methods agree on the hydrological activity of glaciers throughout the winter in both B and Duke watersheds. The Duke Glacier is a polythermal glacier (Flowers et al., 2014; Wilson et al., 2013), and the formation of icings has been commonly
associated with this type of glacier in the Canadian Arctic (Moorman & Michel, 2000; Wainstein et al., 2008), Greenland (Yde & Knudsen, 2005) and Svalbard (Bukowska-Jania & Szafraniec, 2005; Hambrey, 1984; Sobota, 2016; Stachnik et al., 2016; Wadham et al., 200; Yde et al., 2012). The contribution of the Duke Glacier to icing formation detected in the present study confirms that polythermal glaciers are hydrologically active during the winter. Glacier B is also active during winter, but its thermal regime had not yet been categorized. The contribution to icing formation from tributaries connected to small glaciers
situated on valley slopes was not expected because those glaciers are visually thin and therefore probably freeze to their base (Wilson et al., 2013). The exact nature of the discharge in these tributaries should be thus further studied to better understand tributaries' winter activity.

Winter hydrological activity of non-glacier-fed hillslope tributaries has been identified at more than half of the icing locations. This highlights the role of groundwater discharge in the winter hydrological system, even within the proglacial field,



which could have been anticipated to be hydrologically inactive during the winter (Cooper, Hodgkins, Wadham, & Tranter, 2011).

     Another type of hydrological activity in studied watersheds is related to the presence of permafrost. In Alaska and the Canadian Arctic, most of the studied icings have been shown to be fed by subpermafrost water (Hu & Pollard, 1997; Kane & Slaughter, 1973; Kane, 1981; Pollard, 2005; Yoshikawa et al., 2007), and several studies that addressed water provenance for

icing formation in regions with continuous and discontinuous permafrost explicitly concluded that suprapermafrost water does not participate in icing formation (Kane, 1981, in central Alaska, USA; Veillette & Thomas, 1979, in NWT, Canada; and Yoshikawa et al., 2007, in Brooks Range, Alaska, USA). In this study, the results summary provided in Table 2 does not allow us to make a conclusion about the role of suprapermafrost layer water in icings' growth. However, DOC results taken independently can shed some light. In particular, samples from alpine meadow Icings 1 and 2 exhibit higher DOC

concentrations. Among potential sources, suprapermafrost ponds, groundwater wells and one hillslope tributary also exhibit DOC concentrations above the threshold. Interestingly, this hillslope tributary was sampled in the vicinity of the suprapermafrost ponds, suggesting that this tributary is at least partly fed by the suprapermafrost layer water. Potential contribution of suprapermafrost layer water to icing formation is also supported by the analysis of the cryogenic precipitate. As previously observed, icings fed by suprapermafrost water typically stop growing before the end of the winter when the

water stored in the suprapermafrost layer is exhausted (e.g., Pollard, 2005). In the absence of TLC covering alpine meadows, it is impossible to say if discharge from suprapermafrost layer happens throughout the winter or only at the beginning of the winter season.

     While suggested by some methods, the activity of hydrological sources such as buried ice formations and moraine lakes for Icing PF and Icing 4 is impossible to confirm in the absence of DOC in samples from these groups as well as of

cryogenic precipitates on the icings.

     The multi-source and distributed hydrological system described above will potentially respond to a further warming climate by sustaining the increase in winter baseflow in the Upper Duke River watershed. The contribution of small glaciers to winter baseflow, supported by our results, should lead to a future increase in winter baseflow. Liljedahl et al. (2016) suggested for Tanana River, Alaska, that ubiquitous glacier mass loss leads to a continuous supply of meltwater, which is

partly used for aquifer recharge. At the Upper Duke, this recharge could enhance permafrost thaw by heat advection, thus simultaneously increasing aquifer storage capacity (Lamontagne-Hallé et al., 2018; McKenzie & Voss, 2013). As a result, actual glacier retreat should lead to an increase in groundwater discharge during the winter season.

     An increase in suprapermafrost layer thickness in response to the changing climate should lead to aquifers' increased storage capacity, which in turn can also lead to increased winter discharge (Ge et al., 2011; Toohey et al., 2016). In addition,

the formation of taliks activates the groundwater system, which is otherwise shut down during the winter season (Lamontagne-Hallé et al., 2018). Our results show that suprapermafrost layer water is a probable source of icing formation, and therefore it is reasonable to suggest that the observed permafrost thaw in the region (Smith et al., 2015) can contribute to the increased winter discharge in the region.





Similarly, under the increasing temperatures aquifers' water storage capacity can potentially be increased by delayed
freeze-up (Rennermalm et al., 2010; Yang et al., 2002), and thus groundwater-fed hillslope tributaries can be expected to
contribute more to the winter baseflow.

Finally, the moraine lake contribution to winter discharge will most likely increase in a warming climate. Moorman
(2003) and Wainstein et al. (2014) in Bylot Island, Canada, concluded that marginal glacier lakes did not freeze entirely over
the winter and kept releasing water all year long through taliks. The increase in talik formations expected in response to the
changing climate should thus enhance the contribution of moraine lakes to winter discharge.

Based on our results and existing literature, we can anticipate that most of the sources that we detected as hydrologically
active during the winter and contributing to baseflow should exhibit positive feedback to climate change in the Upper Duke
River valley.

## 7 Conclusions

Icings form from hydrological sources that remain active during the winter and therefore have been used as chroniclers of
winter hydrological processes in the Upper Duke River valley, a remote subarctic watershed. The use of TLC images coupled
with the natural-tracers-based analysis of water and icing samples taken at the end of the winter season revealed valuable
information that can be used to understand complex hydrological systems. Samples were exploited by comparing icings'
meltwater isotopic (a), ionic (c), and DOC concentrations to those of potential hydrological sources. In addition, compositions
of solid samples taken on icing remnants (d) were compared to those of potential parent sources.

The results show that it is possible to identify hydrological sources that remain active during the winter by studying
icings formation, and depict a complex, distributed contribution of different sources to studied icings. As expected, icings
forming next to a main glacier terminus are fed by glacier meltwater. However, even in the vicinity of the glaciers, traces of
non-glacier-related water sources, such as groundwater-fed hillslope tributaries, buried ice formations, and moraine lakes, have
been found. As we move downstream from the main glaciers, contribution from groundwater-fed and small-glacier-fed
tributaries is more pronounced, and becomes a major source for icing formation in the alpine meadow section of the watershed.
Some hillslope tributaries within the alpine meadow seem to be fed by suprapermafrost layer water.

Even though Watershed B is a small, highly glacierized catchment, it seems that the sources contributing to icing
formation, and thus the winter hydrological processes occurring there, are comparable to those present in the longer and wider
Upper Duke watershed. However, the Duke watershed shows a larger variety of sources as potential components of winter
runoff. The comparison between Watershed B and the Duke watershed shows that the size of the watershed does not affect the
spatial patterns of winter runoff generation, but a larger watershed is more complex, and its sources are therefore more diverse.

In the context of ubiquitous increase in winter discharge in cold regions, our results show that icing formations can help
overcome the lack of direct observations in these remote environments and provide new insights in winter runoff generation.



The multi-technique approach used in this study provided important information about the water sources active during the winter season in the headwaters of glacierized catchments.

**Acknowledgements**

This research is supported by the Geochemistry and Geodynamics Research Centre (GEOTOP) of Quebec, the Natural Science and Engineering Research Council (NSERC) of Canada, *École de technologie supèrieure*, a constituent of the *Université de*
*Quebec* network, and the Polar Continental Shelf Program. We want to acknowledge that this research was conducted on the traditional territory of Kluane First Nation, and we are thankful for their support. We are also grateful for help from Parks Canada.

**Data availability statement**

The data that support the findings of this study are available from the corresponding author upon reasonable request.

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





**Table 1. Summary of samples collected in (a) Watershed B and (b) Duke watershed. PF stands for proglacial field and AM stands for alpine meadow**

| Sample type | | Sample location | # of samples |
|---|---|---|---|
| **a) Watershed B** | | | |
| Main glacier | PF | At the outlet of the immediate proglacial field where the whole drainage system of the glacier merges into one stream | 1 |
| Main stream | | Same as Main glacier for Watershed B | |
| Icing PF | PF | From supra-icing pond | 1 |
| Buried ice formation | PF | From purged ice wells made by steam-drill | 2 |
| Moraine lake | PF | From lake situated on the left-side ice-cored moraine | 1 |
| Hillslope tributary | PF | Close to the outlet of small streamlets | 5 |
| Glacier-fed tributary | PF | Large tributary formed by water from other glaciers in Watershed B | 1 |
| Icing B | AM | From supra-icing pond | 1 |
| Solids | AM | At Icing B's surface | 1 |
| **b) Upper Duke River watershed** | | | |
| Main glacier | PF | At glacier snout | 4 |
| Main stream | | At the PF/AM delimitation | 3 |
| Icing 4 | PF | From supra-icing pond | 2 |
| | | From water dripping from the icing | 2 |
| Icing 3 | PF | From supra-icing pond | 1 |
| Buried ice formation | PF | From purged ice wells made by steam-drill | 3 |
| | | From meltwater flowing on buried ice–debris cover interface | 2 |
| PFL tributary | PF | From small tributaries, headwaters of which are within the proglacial field | 13 |
| Moraine lake | PF | From kettle moraine lakes | 5 |
| Glacier-fed tributary | PF/AM | At the outlet stream close to glacier snout | 6 |
| | | At the outlet stream close to the confluence with Duke River | 5 |
| Hillslope tributary | PF/AM | From small streamlets, headwaters of which are on the valley hillslopes | 16 |
| Icing 2 | AM | From supra-icing pond | 2 |
| Icing 1 | AM | From supra-icing pond | 2 |
| | | From water dripping from the icing | 1 |
| Suprapermafrost pond | AM | From small ponds underlined by permafrost in the alpine meadow | 3 |
| Groundwater well | AM | From purged shallow (0.5 m) wells | 2 |
| Solid | PF/AM | On Icing 1 and Icing 3's surfaces | 2 |
| Rain | AM | Liquid precipitation | 2 |
| Snow | AM | Taken from snow on the ground | 2 |



**Table 2. Summary of identified potential contributors to icings' formation. (+) indicates the source is supported as contributor to the icing formation while (-) indicates the source is suggested as a non-contributor. Grey cells indicate cases where a particular method is not applicable. Last column shows the overall score for each possible source, where for each icing the maximum positive and negative marks for a given icing are provided to facilitate the interpretation of scores**

| Possible contributor to icing formation | Time lapse images | Stable water isotopes | Ionic signature | Dissolved organic carbon | Solid samples | Overall score |
|---|---|---|---|---|---|---|
| **Icing PF** | | | | | | **+++/--** |
| Main glacier | ++ | | | | | ++ |
| Buried ice formation | + | - | | | | 0 |
| Moraine lake | + | | | | | + |
| **Icing B** | | | | | | **+/--** |
| Glacier-fed tributary | | - | | | - | -- |
| Main stream/glacier B | | - | | | - | -- |
| Hillslope tributary | | | + | | | + |
| Groundwater well | | - | | | - | -- |
| **Icing 1** | | | | | | **+/---** |
| Glacier-fed tributary | | | | - | - | -- |
| Main stream | | | | - | - | -- |
| Hillslope tributary | | | | | | 0 |
| Suprapermafrost pond | | - | | | | - |
| Groundwater well | | | | | - | - |
| **Icing 2** | | | | | | **+/--** |
| Glacier-fed tributary | | | | - | | - |
| Main stream | | - | | - | | -- |
| Hillslope tributary | | | + | | | + |
| Suprapermafrost pond | | - | | | | - |
| Groundwater well | | - | | | | - |
| **Icing 3** | | | | | | **+++/--** |
| Main glacier | | - | | | - | -- |
| Buried ice formation | | - | + | | - | - |
| PFL tributaries | | | | | | 0 |
| Moraine lake | | - | + | | | 0 |
| Glacier-fed tributary | ++ | | | | | ++ |
| Hillslope tributary | ++ | | | | | ++ |
| **Icing 4** | | | | | | **+++/-** |
| Main glacier | ++ | | | | | ++ |
| Buried ice formation | | | + | | | + |
| PFL tributaries | | | | | | 0 |
| Moraine lake | | - | | | | - |
| Glacier-fed tributary | ++ | | | | | ++ |
| Hillslope tributary | ++ | | | | | ++ |



Figure 1. Study watersheds and sampling plans. Panel (a) shows Upper Duke River watershed and Watershed B limits and the locations of time lapse cameras and the automatic weather station. Panels (b) and (c) present schematic maps of Watershed B and Duke watershed, respectively, with relative locations of sampling points. Dashed lines show sources that are not directly connected above the surface with the main stream. In panel (c), for streams fed by small glaciers, "a" identifies samples taken close to the glacier tongue, and "b" identifies samples taken close to the confluence with the Duke River.



**Figure 2.** Glacier B tongue and hydrological activities visible in the TLC1 images. Panel (a) shows Glacier B's tongue, proglacial field, proglacial icing, and the right side of the valley at the camera installation. Panels (b) and (c) illustrate flooding events on the left (b) and the right (c) side of the proglacial field; panels (d) and (e) show flooding occurring on the left bottom side of the proglacial field.



**Figure 3. The Duke proglacial field visible in the TLC3 images. Panel (a) shows Icings 3 (right side of the image) and 4 (left side of the image) at the camera installation; panel (b) shows an example of flooding originating from the Duke Glacier; panels (c) and (d) show an example of the darkening/bluing of pixels as a result of flooding; panels (e) and (f) show an example of flooding being detectable from reflective (brightening) pixels.**



**Figure 4. Hydrological activities observed in Watershed B by the use of time lapse cameras for the 2015–2016 season (top panel) and the 2016–2017 season (bottom panel). Black solid lines show air temperature measured at the automatic weather station; half-circles mark the timing of hydrological activity on the left/right side of the proglacial field, and full circles represent the timing of hydrological activity related to either an ice-cored moraine or a moraine lake.**







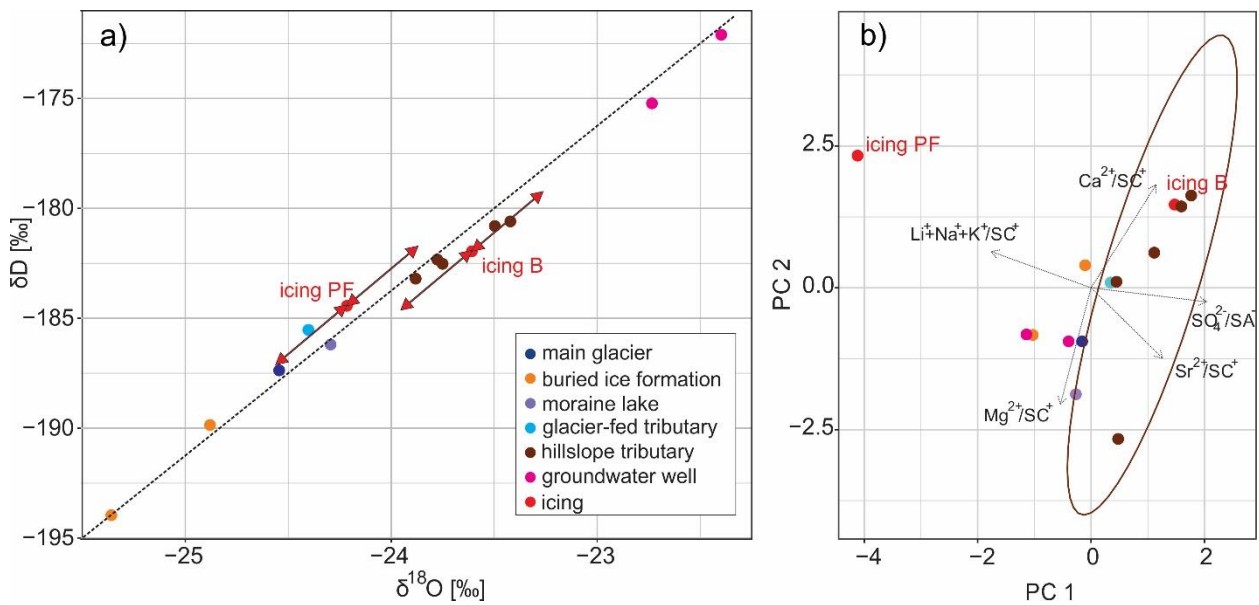

**Figure 5. Isotopic and hydrochemical methods result for Watershed B. Panel (a) shows results from stable water isotope analysis. The straight dashed line represents the Local Meteoric Water Line. Red arrows define the range within which samples from parent sources are considered to plot close to the icing sample. Panel (b) shows results of hydrochemical analysis. Ellipse illustrates 95%**
**confidence interval.**





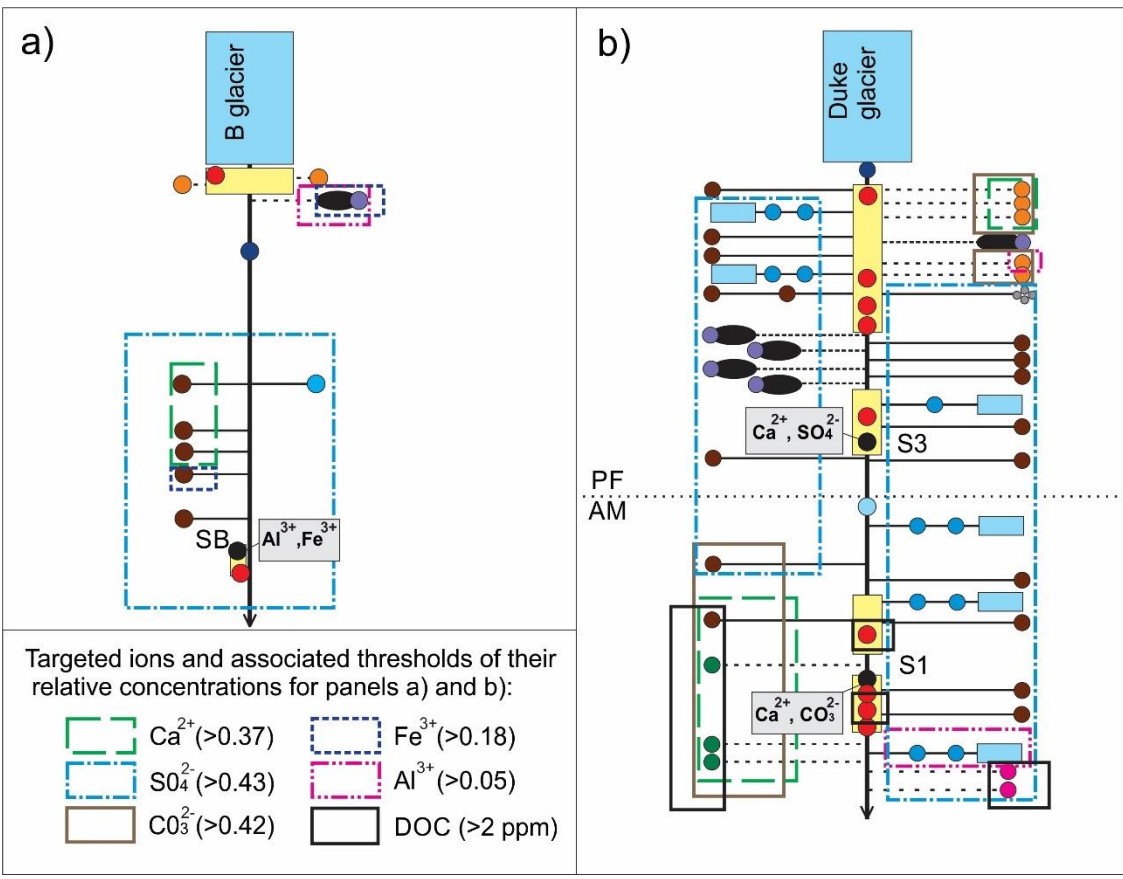

**Figure 6. Conceptual maps of ionic content of water samples. Colored boxes delineate sampled areas with targeted ion relative concentration above the threshold. The names of sampling points can be found in Figure 1.**








**Figure 7. Hydrological activities observed in the Upper Duke watershed by the use of time lapse cameras for the 2015–2016 season (top panel) and the 2016–2017 season (bottom panel). Black solid lines show air temperature measured at the automatic weather station. Circles represent the timing of hydrological activity related to different water sources. The circle color identifies a specific**
**source type. In cases where different sources were observed as being active on the same day, the circle has multiple colors.**





**Figure 8. Isotopic and hydrochemical results for the Duke watershed. Panels (a) and (b) show icings and sources within the proglacial field; panels (c) and (d) show icings and sources within the alpine meadow. Panels (a) and (c) show results from stable water isotope analysis. The straight dashed line represents the Local Meteoric Water Line. Red arrows define the range within which samples from parent sources are considered to plot close to the icing sample. For samples from Icing 4, only arrows for the most depleted and the less depleted samples are included for readability. Panels (b) and (d) show results of hydrochemical analysis. Ellipse illustrates 95% confidence interval.**