# Peer review of "Proglacial icings as records of winter hydrological processes"

_The Cryosphere, 2020_

## Referee Comment (RC1) · Olga Makarieva (Referee) · 11 Jul 2020

Olga Makarieva (Referee)

omakarieva@gmail.com

Aufeis studies have been recently getting second breath in the Arctic countries and this tendency should be fully supported. Aufeis are visible manifestation of complicated hydrological, geocryological and hydrogeological processes in the cold regions and their changes in warming climate. The new approaches of their investigation that may provide additional information on the sources of winter flow and their transformation are highly relevant for understanding of changing hydrological regime. Chesnokova et al. proposed the combination of direct (time lapse cameras) and indirect (chemical) methods to study the sources of aufeis fields in the glacierized catchments in the Yukon River basin. The analysis of visible images of aufeis formation during winter season supported by the analysis of water and ice samples on stable water isotopes, ionic

composition, DOC allowed for qualitative distinguishing between the potential sources of aufeis feeding. Some simple adjustments like using the ice rails in the view fields of the time lapse cameras, assessment of aufeis fields areas and volumes at the end of spring by direct measurements and remote sensing data analysis, precipitation measurements would add qualitative value to the study. In general, the paper is well written, the explanations are clear and descriptive. It is important that the authors are aware and discuss the limitations of the proposed approach and obtained results. General recommendation to the authors would be not to limit the literature review to North American authors and pay attention to the Russian studies of recent hydrological changes and aufeis phenomena, all the more so that the first author clearly is able to read in Russian. Specific comments Line 17-19: I would suggest eliminating the following sentence from the abstract "If confirmed in other cold regions, those results will suggest orienting winter flow trend studies toward a multi-causal hypothesis in glacierized catchments", as it sounds a bit boastfully. Line 445: change naturel to natural

Last comment and scientific regards to the authors from the field studies of giant aufeis in the North-East of Russia (the Magadan region) where we are at the moment using the same methods as proposed in the paper – TLC, water and ice samples, though complimented by hydrological measurements of aufeis input into river streamflow. The photo is attached.
* * *
[Figure]

**Fig. 1.**

---

## Referee Comment (RC2) · Tim Ensom (Referee) · 14 Jul 2020

1) Overall Quality

In my opinion the manuscript by Chesnokova et al. is a valuable contribution to mountain and winter hydrology, and to general methodology for winter hydrological investigations in cold regions. I think it is a good fit for The Cryosphere. The title and abstract represent the manuscript well. The work includes a substantial review primarily of North American literature in its Introduction and Discussion. The purpose of the work as I understand – to improve understanding of winter hydrological connectivity in a glaciated mountain catchment in order to improve projections of hydrological response with continued climate change – is well articulated. While the approach of geochemical

sampling and analysis of aufeis is not new, to my knowledge there are few examples in the literature of this approach in high-altitude glaciated catchments.

There is an opportunity for greater detail in the description of how samples were obtained from the icings and the extent to which they might represent all ice layers. I think that the thresholds for classifying water sources as contributing to the icings or not, on the basis of the distance between their samples and those of the icings in co-isotope plots, might be too restrictive and not fully consider isotope fractionation. The authors might be able to expand on the rationale for this approach, as it could be based on isotope work that I am not aware of. I think that the conclusions are substantial in relation to the type of hydrological system being investigated, and can help guide the design of catchment-scale studies in other cold regions.

The specific comments and technical suggestions below provide ideas for minor improvements in language.

2) Specific Comments

Line 124: Would it be possible to replace the term "drastically" with a word that would quantify how limited the changes in chemical and isotopic properties can be, based on the Beylich and Laute reference?

Line 142: It would be helpful for me if the opposite principal was briefly described. I think the opposite principal might be that similarities in the composition of an icing and a potential water source according to multiple tracers would suggest that the water source was indeed parent water for the icing, but I could be incorrect.

Line 295: I don't have extensive experience in isotope geochemistry, but it seems to me that there could be a greater separation in co-isotope space (in the 18O vs. D figure) between an icing sample and its source than the distance between Glacier B and Icing PF. Because of cryogenic fractionation, if Icing PF was a closed system icing it could be up to about 2.75 per mil 18O higher than the source glacier. Are the samples of Icing

PF expected to represent each icing layer and provide an approximate mean value for the d18O of the icing? The 2011 paper by Lacelle ( DOI: 10.1002/ppp.712 ) might be helpful, and seeing it has also helped me in my work.

Line 329: I can't distinguish these events/activities based on looking at the top panel of Figure 7. All blue points are identified as Duke Glacier. Could labels be added?

Section 5.2: I'm not sure that the distance obtained in a co-isotope plot for watershed B has really been established well as the basis for ruling out potential sources.

Section 5.3: I find this analysis more convincing than the approach with isotopes, due to the uncertainty I have over the fixed distance between the icing sample and potential source considered to be necessary in the isotope diagrams. I may have misunderstood part of the rationale for applying the fixed distance in co-isotope space as a threshold.

Section 6.1.2: I think that the indirect methods actually provide the most convincing evidence of the sources for the icings, and that the TLC images mainly provide information about the timing of icing growth.

Lines 472-487 (paragraph): Interesting and informative discussion.

3) Technical Suggestions

Line 39: Would it be correct to say "trends in winter discharge are observed in areas with and without permafrost"?

Line 43: Suggest saying "in cold regions, testing the hypothesis about" Lines 84, 85: Does the reference also provide ranges for mean annual air temperature and mean annual total precipitation? If a range is given instead of a single value for a mean, it might make readers wonder if there is more information to explain what the lower and higher numbers refer to.

Line 94: Arctic should be capitalized.

Line 104: "as it flows downstream" could be removed if the manuscript needs to be

shortened.

Line 105: Consider saying "two areas", consider removing "actual"

Line 107: Is "pronounced" used to refer to steep? If so, it might be more clear to say "less steep, vegetated slopes".

Line 112: I suggest "icing formation" instead of "icings formation", with icing being plural in this case.

Line 115: Suggest "TLC coverage, however, is limited in its range, is vulnerable"

Line 130: Would it be better to say "The same phenomenon can affect the relation"

Line 132: "because there is a link between stable"

Line 135: Can you clarify what limits are being referred to?

Line 139: It's still not clear to me what limitations have been mentioned recently in the text. Is the limitation the fact that icings can't contain things that their source waters don't contain?

Line 141: Would it be better to say "given source for a unique tracer do not mean that the source is a parent water"

Line 141: I suggest saying "between icings" instead of "between icing"

Line 149: I suggest "...oriented to target icing formations..."

Line 163: "southwest shore" might be more descriptive than "right shore".

Line 166: I suggest: "During the field campaign of June 2016 four icing remnants within the Duke riverbed (Figure 1a and c) and two in Watershed B (Figure 1a and b) were observed and sampled". This removes the first person from the sentence, and is conventional for describing methods.

Line 172-173: Do you mean bubble-rich? Line 190: It may help readers if locations

were described using east/west instead of right and left. There are a few other instances of left/right in the paper that might be reviewed.

Line 199: Might be better to state "Rainwater samples were also collected during the field campaign."

Line 226: "We measured the anion concentrations in the samples"

Line 276: Suggest "of November, icing formation was observed on the (west/east/north/south) side of the proglacial field."

Line 280: Comma may not be needed after "16 February"

Line 302: Could a definition be included of SC+ and SA- ? If I understand correctly these would be the sum of cations or sum of anions.

Line 316: Could say instead: "The sampled solids are therefore most likely sediments carried out by parent waters."

Line 318: I think it would be fine to remove the portion of the sentence "On the other hand".

Line 330: I suggest "appear to be contributing". Can there be a more detailed description of the evidence that suggests that they are contributing? What was observed in the TLC images?

Line 367-370: This sentence might benefit from being broken into two. A reference might be important to include if a statement is being made that high DOC has been associated with suprapermafrost water in other research.

The heading beside 6.1 could be changed to "Icing formation parent source identification"

Line 394: I suggest "water sources for each of the icings that were in the cameras' fields of vision." Line 396-397: Might be better to say "contribute to the formation of

Icings 3 and 4."

Line 401: Perhaps say systems instead of set-ups.

Line 406: Check parentheses and adjust or add comma.

The sentence starting with "These results", spanning lines 423 to 425, isn't clear to me.

Line 440. Might be good to change "as" to ", and" after the word watersheds, but just a suggestion. The use of indirect methods would have been informative (in my opinion) even if they had contradicted TLC findings.

Line 454: responsible for icing formation (leave out the s)

Line 515: I suggest saying "have been used to chronicle winter hydrological"

Line 517: I suggest saying "natural-tracer-based analysis"

Line 518: Could say that samples were analysed/investigated/explored. "exploited" doesn't seem like the correct term

---

## Author Comment (AC2) · 11 Sep 2020

The comment was uploaded in the form of a supplement:
https://tc.copernicus.org/preprints/tc-2020-63/tc-2020-63-AC2-supplement.pdf

———————————

---

## Author Response (AR1)

September, 2020

Dear Dr. Morse,

Thank you for the feedback on our manuscript "Proglacial icings as records of winter hydrological processes" [Paper tc-2020-63] submitted for publication in *Cryosphere*. We greatly appreciated the comments from the two reviewers, Olga Makarieva and Tim Ensom. Based on these comments we have made substantial changes to the manuscript and believe that it is significantly enhanced as a result. Included with our resubmission is the manuscript with all changes highlighted in yellow. Point-by-point replies to each of the referees' comments follow here under. Comments are in italics and replies are in normal font. Furthermore, to track the changes in the revised version of the manuscript, their location is indicated in bold. The notation used to locate the changes first defines the page number, then the line number(s). For example, **P4L15** means that the described modification to the manuscript can be found on the 15th line of the 4th page in the revised version (i.e. the version with changes highlighted in yellow).

The manuscript presents important new insights into the winter baseflow production in two highly glacierized watersheds of different size in the southwestern Yukon. The method that was developed for this research allowed identifying several hydrological sources contributing to the formation of icing within both proglacial field and alpine meadow parts of these watersheds. Our results show that, in addition to main glaciers, hydrological sources such as hillslope tributaries, possibly fed by suprapermafrost layer water, contribute to icing growth during winter and thus to winter discharge. If confirmed in other cold regions, those results will suggest orienting winter flow trend studies toward multi-causal hypothesis in glacierized catchments. More generally, this study shows the potential of using icing formations as a new, barely explored source of information on cold regions winter hydrological processes that can contribute to overcoming the paucity of observations in these regions.

Thank you for your consideration and we look forward to your response.
Sincerely,

Anna Chesnokova, Michel Baraër and Émilie Bouchard

Construction Engineering Department
École de technologie supérieure
Montreal, QC, Canada

**Reviewer #1**

Thank you for the feedback on our manuscript! Replies for specific comments are here under (comments are in italics and replies are in normal font).

*General recommendation to the authors would be not to limit the literature review to North American authors and pay attention to the Russian studies of recent hydrological changes and aufeis phenomena, all the more so that the first author clearly is able to read in Russian.*

We really appreciate your suggestion. We now included the following studies published by Russian researchers on winter river discharge increase as well as icings formation (they are highlighted in the text):

Alekseyev, V.R.: Naledi. Novosibirsk, Nauka, Moscow, 1987 (in Russian).

Alekseyev, V.R.: Cryogenesis and geodynamics of icing valleys. Geodynamics & 366 Tectonophysics 6 (2), 171–224, 2015. doi:10.5800/GT-2015-6-2-0177.

Makarieva, O., Nesterova, N., Andrew Post, D., Sherstyukov, A., & Lebedeva, L. (2019). Warming temperatures are impacting the hydrometeorological regime of Russian rivers in the zone of continuous permafrost. *Cryosphere*, *13*(6), 1635–1659. https://doi.org/10.5194/tc-13-1635-2019

Markov ML, Vasilenko NG, Gurevich EV. Icing fields of the BAM zone: expeditionary investigations. Saint Petersburg, Russia: Nestor-History; 2017.

Pomortsev OA, Kashkarov EP, Popov VF. Aufeis: global warming and processes of ice formation (rhythmic basis of long-term prognosis). Yakutsk State Univ Bull. 2010;7:40-48.

Tananaev, N. I., Makarieva, O. M., and Lebedeva, L. S.: Trends in annual and extreme 437 flows in the Lena River basin, Northern Eurasia. doi: 10.1002/2016GL070796, 2016.

*Specific comments Line 17-19: I would suggest eliminating the following sentence from the abstract "If confirmed in other cold regions, those results will suggest orienting winter flow trend studies toward a multi-causal hypothesis in glacierized catchments", as it sounds a bit boastfully.*

We modified the sentence to "If confirmed in other cold regions, those results could confirm the multi-causal hypothesis of winter discharge increase in glacierized catchments". This wording allows us to highlight that our study supports the multi-causal hypothesis for increase in winter flows without sounding boastful as you pointed out.

*Line 445: change naturel to natural*

Thank you, it has been fixed!

*Last comment and scientific regards to the authors from the field studies of giant aufeis in the North-East of Russia (the Magadan region) where we are at the moment using the same methods as proposed in the paper – TLC, water and ice samples, though complimented by hydrological measurements of aufeis input into river streamflow. The photo is attached.*

Thank you for sharing the picture, and we hope that your field investigation was successful!

**Reviewer #2**

We greatly appreciated your thorough and constructive review!

Point-by-point replies to each of the comments follow.

*1) General comments*

*In my opinion the manuscript by Chesnokova et al. is a valuable contribution to mountain and winter hydrology, and to general methodology for winter hydrological investigations in cold regions. I think it is a good fit for The Cryosphere. The title and abstract represent the manuscript well. The work includes a substantial review primarily of North American literature in its Introduction and Discussion. The purpose of the work as I understand – to improve understanding of winter hydrological connectivity in a glaciated mountain catchment in order to improve projections of hydrological response with continued climate change – is well articulated. While the approach of geochemical sampling and analysis of aufeis is not new, to my knowledge there are few examples in the literature of this approach in high-altitude glaciated catchments.*

*There is an opportunity for greater detail in the description of how samples were obtained from the icings and the extent to which they might represent all ice layers.*

This is indeed an important question. The sampling took place during the summer season and icings were not sampled from each layer, but from the meltwater dripping from them. By doing so the goal was to sample icing's average properties. We acknowledge that sampling each layer independently would have provided a more comprehensive picture of the icing hydrochemical signature. Unfortunately, our goal being providing a primary evaluation of different methods for linking icings characteristics to contributing sources, sampling and analyzing each layer of ice was not necessary to meet this study's objectives and would have been logistically more complex for both sampling and data analysis.

Those limits have been taken into consideration at the data treatment and methods assessment stage as described in the section 3.1 of the manuscript.

To make sure no ambiguities are present in the manuscript on this matter, we have now added "and about icing samples representativity" in the discussion about the methodological limits of the method (**P15L452**). We also added "Thus, most of the icing samples represent an average signature of all the icing layers" where we discuss the sampling methodology (**P6L180-181**).

*I think that the thresholds for classifying water sources as contributing to the icings or not, on the basis of the distance between their samples and those of the icings in co-isotope plots, might be too restrictive and not fully consider isotope fractionation. The authors might be able to expand on the rationale for this approach, as it could be based on isotope work that I am not aware of.*

The threshold applied for identifying potential non-contributing sources to icings formation aims to account for limits in sampling technics and in sample representativeness (Section 6.2 of the manuscript). The goal of rejecting as potential contributor all samples which are not within the thresholds (as an opposite to considering those within the thresholds as potential contributors) is to avoid mis-identifying potential contributing sources. This technic therefore accounts for all kind of uncertainties, including those arising from possible isotopic fractionation. More specific modifications made in the manuscript to clarify this point are described below in the specific comments section.

*I think that the conclusions are substantial in relation to the type of hydrological system being investigated, and can help guide the design of catchment-scale studies in other cold regions.*

Thank you, we appreciate the encouraging comment!

*The specific comments and technical suggestions below provide ideas for minor improvements in language.*

*2) Specific Comments*

*Line 124: Would it be possible to replace the term "drastically" with a word that would quantify how limited the changes in chemical and isotopic properties can be, based on the Beylich and Laute reference?*

Thank you for this suggestion. In Beylich and Laute (2012), the authors do not really quantify this change, and refer to their Figures 2 and 5 when saying "the relative ionic composition of the surface water (see also Fig. 2) is not significantly modified and shows no obvious intra-annual temporal variations (Fig. 5)". These figures show relative ionic compositions, but, unfortunately, do not show the exact percentage of each component.

To address this comment, we changed "water chemical and isotopic properties" to "relative chemical and isotopic concentrations" and used word "significantly" instead of "drastically" as they use in the paper (**P4L125**).

*Line 142: It would be helpful for me if the opposite principal was briefly described. I*

*think the opposite principal might be that similarities in the composition of an icing and a potential water source according to multiple tracers would suggest that the water source was indeed parent water for the icing, but I could be incorrect.*

We regret the lack of clarity in our initial manuscript. Because multiple tracers "agree" with each other in the PCA diagrams (unlike for isotopes or DOC where only one or two tracers agree), we can be more confident that this indeed indicates that icing and parent sources are related.

To clarify, we changed the sentence to "The opposite principle (i.e., the similitude between an icing and a given sources  implies that the source is a potential parent water) applies where results of many tracers agree with each other, as it is the case for the ionic signature-based PCA" (**P5L143-145**).

*Line 295: I don't have extensive experience in isotope geochemistry, but it seems to me that there could be a greater separation in co-isotope space (in the 18O vs. D figure) between an icing sample and its source than the distance between Glacier B and Icing PF. Because of cryogenic fractionation, if Icing PF was a closed system icing it could be up to about 2.75 per mil 18O higher than the source glacier. Are the samples of Icing PF expected to represent each icing layer and provide an approximate mean value for the d18O of the icing? The 2011 paper by Lacelle ( DOI: 10.1002/ppp.712 ) might be helpful, and seeing it has also helped me in my work.*

We are very grateful for the pointed-out reference; it is a very useful and interesting study. Our rational for the selected threshold is based on the fact that we do not know precisely what is happening in winter in the valley. The only thing we can observed from TLC is that there is an outflow from the glacier tongue onto the icing surface. It is not possible to say if freezing was happening in open or closed system. However, we are confident that the B glacier feeds the icing PF based on TLC images. Thus, we arbitrarily chose the distance between the B Glacier and Icing PF samples on the isotopes diagram as being "near".

To address this issue and clarify this choice, we added a sentence in the manuscript (**P10L298-301**).

*Line 329: I can't distinguish these events/activities based on looking at the top panel of Figure 7. All blue points are identified as Duke Glacier. Could labels be added?*

Thank you for this comment, it gave us an opportunity to spot an error on the **P11L335**: Instead of "GL1", we meant "GL2".

Dark blue points show the activity of the Duke glacier, while light blue points show the activity of glacier-fed tributaries. Originally, we specified on those diagrams all the detected sources. However, since the goal of the study was identifying groups of sources responsible for icing formation, we thought it would be less overwhelming if we only keep differentiating on these

Figures (4 and 7) between the main glacier, glacier-fed tributaries, hillslope tributaries, etc. (as we do in other figures and in Table 2), instead of labeling each particular sources.

We change the "visible" to "detected" to clarify that it was detected by TLC, but not visible per se on the panel (**P11L335**).

*Section 5.2: I'm not sure that the distance obtained in a co-isotope plot for watershed B has really been established well as the basis for ruling out potential sources.*

Please, see our reply above (for the line 295)

*Section 5.3: I find this analysis more convincing than the approach with isotopes, due to the uncertainty I have over the fixed distance between the icing sample and potential source considered to be necessary in the isotope diagrams. I may have misunderstood part of the rationale for applying the fixed distance in co-isotope space as a threshold.*

We had to apply different methods for isotopes than for PCA since multiple tracers are used for PCA, and, as a result, we can be more confident when they agree with each other. For isotopes, in order to avoid false positive results, we opted for a stricter methodology. We clarified it in the methods overview section (**P5L143-145**).

*Section 6.1.2: I think that the indirect methods actually provide the most convincing evidence of the sources for the icings, and that the TLC images mainly provide information about the timing of icing growth.*

The problem of using only indirect methods is that, without direct observations, multiple hypotheses explaining the observed results may exist. Each method has its own limitations. TLC images allow us to either limit the number of possible solutions or to confirm the results from chemical anaysis.

*Lines 472-487 (paragraph): Interesting and informative discussion.*

Thank you!

*3) Technical Suggestions*

*Line 39: Would it be correct to say "trends in winter discharge are observed in areas with and without permafrost"?*

Yes, but we prefer to keep the term "permafrost-underlain" to be consistent with the cited references.

*Line 43: Suggest saying "in cold regions, testing the hypothesis about"*

Thank you for the suggestion. The text has been modified accordingly.

*Lines 84, 85: Does the reference also provide ranges for mean annual air temperature and mean annual total precipitation? If a range is given instead of a single value for a mean, it might make readers wonder if there is more information to explain what the lower and higher numbers refer to.*

Thank you for pointing this out. Actually, we provided here a range for a larger region (Southerwestern Yukon). Upon rereading we realized that the original sentence was rather imprecise, and decided changed this part to "with a mean annual air temperature between −8 and −12°C and a mean annual total precipitation between 500-600 mm depending on the location in the region" (**P3L86-87**)

*Line 94: Arctic should be capitalized.*

Thank you. We capitalized every reference to both the "Arctic" and "Subarctic".

*Line 104: "as it flows downstream" could be removed if the manuscript needs to be shortened.*

Yes, we agree, it has been removed.

*Line 105: Consider saying "two areas", consider removing "actual"*

We removed "actual".

*Line 107: Is "pronounced" used to refer to steep? If so, it might be more clear to say "less steep, vegetated slopes".*

Thank you for pointing this out. We modified the sentence for the one you suggested, as it is clearer.

*Line 112: I suggest "icing formation" instead of "icings formation", with icing being plural in this case.*

As suggested, we changed every reference to "icings" to its singular form when used with "formation".

*Line 115: Suggest "TLC coverage, however, is limited in its range, is vulnerable"*

We applied the suggested modification.

*Line 130: Would it be better to say "The same phenomenon can affect the relation"*

Yes, thank you.

*Line 132: "because there is a link between stable"*

We removed "that exists".

*Line 135: Can you clarify what limits are being referred to?*

We appreciate the comment because it gave us an opportunity to clarify this aspect in the manuscript. We changed "limits" to "aforementioned processes" to refer to the processes that were discussed above, and added "interpretation is limited" to clarify that it is a results of those processes: "Taking into account those aforementioned processes, interpretation of the results from the indirect methods for each potential parent source is limited and is simply expressed either as (+) indicating that the source is supported as a contributor to the icing formation, or as (-) indicating that the source is suggested as not being a contributor" (**P5L135-138**)

*Line 139: It's still not clear to me what limitations have been mentioned recently in the text. Is the limitation the fact that icings can't contain things that their source waters don't contain?*

Yes, see previous comment.

*Line 141: Would it be better to say "given source for a unique tracer do not mean that the source is a parent water"*

We appreciate your comment, but we feel that the expression used in the manuscript and the one suggested are equivalent, so we decided to keep this part as it was.

*Line 141: I suggest saying "between icings" instead of "between icing"*

We refer here to a single icing, which means we forgot to add "the" in the sentence. We modified "between icing and a giving source" to "between the icing and a given source". Thank you for pointing out this mistake.

*Line 149: I suggest "...oriented to target icing formations..."*

We applied this suggestion.

*Line 163: "southwest shore" might be more descriptive than "right shore".*

Thank you, we changed it everywhere in the manuscript.

*Line 166: I suggest: "During the field campaign of June 2016, four icing remnants within the Duke riverbed (Figure 1a and c) and two in Watershed B (Figure 1a and b) were observed and sampled". This removes the first person from the sentence, and is conventional for describing methods.*

Thank you for the comment. However, we opted to use an active voice for this manuscript so we prefer to keep the first-person form.

*Line 172-173: Do you mean bubble-rich?*

Yes! Thank you, we corrected it.

*Line 190: It may help readers if locations were described using east/west instead of right and left. There are a few other instances of left/right in the paper that might be reviewed.*

We think this is a very good idea. We changed it everywhere.

*Line 199: Might be better to state "Rainwater samples were also collected during the field campaign."*

As previously mentioned, we prefer to use an active voice.

*Line 226: "We measured the anion concentrations in the samples"*

Yes, done.

*Line 276: Suggest "of November, icing formation was observed on the (west/east/north/south) side of the proglacial field."*

Thank you for the idea, we changed to "west" everywhere.

*Line 280: Comma may not be needed after "16 February"*

Thank you, we removed it.

*Line 302: Could a definition be included of SC+ and SA- ? If I understand correctly these would be the sum of cations or sum of anions.*

Yes, this is correct. We added those definitions at **P8L244**.

*Line 316: Could say instead: "The sampled solids are therefore most likely sediments carried out by parent waters."*

We modified this sentence based on your suggestion.

*Line 318: I think it would be fine to remove the portion of the sentence "On the other hand".*

We prefer to keep it because it highlights that there is still a possibility (the other possibility) that those waters are actually sources waters.

*Line 330: I suggest "appear to be contributing". Can there be a more detailed description of the evidence that suggests that they are contributing? What was observed in the TLC images?*

Thank you, we modified this sentence. Those contributions were detected by flooding events next to the potential source observed in the TLC images. This was added in the text.

*Line 367-370: This sentence might benefit from being broken into two. A reference might be important to include if a statement is being made that high DOC has been associated with suprapermafrost water in other research.*

We regret the lack of clarity in our initial manuscript. It has been clarified it by decomposing the sentence into a) and b) (**P17L370-371**).

*The heading beside 6.1 could be changed to "Icing formation parent source identification"*

We changed it to "Parent sources identification for icing formation".

*Line 394: I suggest "water sources for each of the icings that were in the cameras' fields of vision."*

It has been modified as suggested.

*Line 396-397: Might be better to say "contribute to the formation of Icings 3 and 4."*

Yes, we applied your suggestion.

*Line 401: Perhaps say systems instead of set-ups.*

It has been added.

*Line 406: Check parentheses and adjust or add comma.*

Thank you, done.

*The sentence starting with "These results", spanning lines 423 to 425, isn't clear to me.*

Sorry for the confusion. We rephrased it to remind the reader that we used fix distance between B Glacier sample and Icing PF from the isotope diagram to identify parent sources for other icings (**P14L429-430**).

*Line 440. Might be good to change "as" to ", and" after the word watersheds, but just a suggestion. The use of indirect methods would have been informative (in my opinion) even if they had contradicted TLC findings.*

Yes, we agree.

*Line 454: responsible for icing formation (leave out the s)*

Thank you, this change has been applied everywhere in the manuscript.

*Line 515: I suggest saying "have been used to chronicle winter hydrological"*

Yes, done (**P17L521**).

*Line 517: I suggest saying "natural-tracer-based analysis"*

Thank you for your suggestion. The sentence has been changed accordingly.

*Line 518: Could say that samples were analysed/investigated/explored. "exploited" doesn't seem like the correct term*

We wrote "explored" instead.

---

## Editor Decision (ED1)

[revised manuscript text omitted]

**Figure 1. Study watersheds and sampling plans. Panel (a) shows the Upper Duke River watershed and Watershed B limits and the locations of time-lapse cameras and the automatic weather station. Panels (b) and (c) present schematic maps of Watershed B and Duke watershed, respectively, with relative locations of sampling points. Dashed lines show sources that are not directly connected above the surface with the main stream. In panel (c), for streams fed by small glaciers, "a" identifies samples taken close to the glacier tongue, and "b" identifies samples taken close to the confluence with the Duke River.**

[Figure]

**Figure 2. Glacier B tongue and hydrological activities visible in the TLC1 images. Panel (a) shows Glacier B's tongue, proglacial field, proglacial icing, and the east side of the valley at the camera installation. Panels (b) and (c) illustrate flooding events on the west (b) and the east (c) side of the proglacial field; panels (d) and (e) show flooding occurring on the east bottom side of the proglacial field.**

795

[Figure]

**Figure 3. The Duke proglacial field visible in the TLC3 images.** Panel (a) shows Icings 3 (right side of the image) and 4 (left side of the image) at the camera installation; panel (b) shows an example of flooding originating from the Duke Glacier; panels (c) and (d) show an example of the darkening/bluing of pixels as a result of flooding; panels (e) and (f) show an example of flooding being detectable from reflective (brightening) pixels.

800

[Figure]

805

**Figure 4. Hydrological activities observed in Watershed B by the use of time-lapse cameras for the 2015–2016 season (top panel) and the 2016–2017 season (bottom panel). Black solid lines show air temperature measured at the automatic weather station; half-circles mark the timing of hydrological activity on the left/right side of the proglacial field, and full circles represent the timing of hydrological activity related to either an ice-cored moraine or a moraine lake.**

[Figure]

**Figure 5. Isotopic and hydrochemical methods result for Watershed B. Panel (a) shows results from stable water isotope analysis. The straight dashed line represents the Local Meteoric Water Line. Red arrows define the range within which samples from parent sources are considered to plot close to the icing sample. Panel (b) shows results of hydrochemical analysis. Ellipse illustrates 95% confidence interval.**

[Figure]

**Figure 6. Conceptual maps of ionic content of water samples. Colored boxes delineate sampled areas with targeted ion relative concentration above the threshold. The names of sampling points can be found in Figure 1.**

[Figure]

**Figure 7. Hydrological activities observed in the Upper Duke watershed by the use of time-lapse cameras for the 2015–2016 season (top panel) and the 2016–2017 season (bottom panel). Black solid lines show air temperature measured at the automatic weather station. Circles represent the timing of hydrological activity related to different water sources. The circle color identifies a specific source type. In cases where different sources were observed as being active on the same day, the circle has multiple colors.**

[Figure]

**Figure 8. Isotopic and hydrochemical results for the Duke watershed. Panels (a) and (b) show icings and sources within the proglacial field; panels (c) and (d) show icings and sources within the alpine meadow. Panels (a) and (c) show results from stable water isotope analysis. The straight dashed line represents the Local Meteoric Water Line. Red arrows define the range within which samples from parent sources are considered to plot close to the icing sample. For samples from Icing 4, only arrows for the most depleted and the less depleted samples are included for readability. Panels (b) and (d) show results of hydrochemical analysis. Ellipse illustrates 95% confidence interval.**

---

## Author Response (AR2)

Dear Dr. Morse,

Thank you for your thorough feedback on our manuscript! We greatly appreciated all the comments, and changed the manuscript accordingly.

Thank you for your consideration and we look forward to your response.
Sincerely,

Anna Chesnokova, Michel Baraër and Émilie Bouchard

Construction Engineering Department
École de technologie supérieure
Montreal, QC, Canada